# ATR inhibitors as a synthetic lethal therapy for tumours deficient in ARID1A

Chris T. Williamson[1,2], Rowan Miller[1,2], Helen N. Pemberton[1,2], Samuel E. Jones[1,2], James Campbell[1,2], Asha Konde[1,2], Nicholas Badham[1,2], Rumana Rafiq[1,2], Rachel Brough[1,2], Aditi Gulati[1,2], Colm J. Ryan[3], Jeff Francis[1,2], Peter B. Vermulen[2,4], Andrew R. Reynolds[2], Philip M. Reaper[5], John R. Pollard[5], Alan Ashworth[1,2,†] & Christopher J. Lord[1,2]

Identifying genetic biomarkers of synthetic lethal drug sensitivity effects provides one approach to the development of targeted cancer therapies. Mutations in *ARID1A* represent one of the most common molecular alterations in human cancer, but therapeutic approaches that target these defects are not yet clinically available. We demonstrate that defects in *ARID1A* sensitize tumour cells to clinical inhibitors of the DNA damage checkpoint kinase, ATR, both *in vitro* and *in vivo*. Mechanistically, ARID1A deficiency results in topoisomerase 2A and cell cycle defects, which cause an increased reliance on ATR checkpoint activity. In *ARID1A* mutant tumour cells, inhibition of ATR triggers premature mitotic entry, genomic instability and apoptosis. The data presented here provide the pre-clinical and mechanistic rationale for assessing ARID1A defects as a biomarker of single-agent ATR inhibitor response and represents a novel synthetic lethal approach to targeting tumour cells.

[1] The CRUK Gene Function Laboratory, The Institute of Cancer Research, London SW3 6JB, UK. [2] The Breast Cancer Now Toby Robins Breast Cancer Research Centre, The Institute of Cancer Research, London SW3 6JB, UK. [3] Systems Biology Ireland, University College Dublin, Dublin 4, Ireland. [4] GZA Hospitals Sint-Augustinus, Wilrijk, Belgium and Center for Oncological Research, University of Antwerp, Oosterveldlaan 24, Wilrijk Antwerp 2610, Belgium. [5] Vertex Pharmaceuticals (Europe) Limited, Milton Park, Abingdon, Oxfordshire OX14 4RY, UK. † Present address: UCSF Helen Diller Family Comprehensive Cancer Centre, San Francisco, California 94158, USA. Correspondence and requests for materials should be addressed to A.A. (email: Alan.Ashworth@ucsf.edu) or to C.J.L. (email: Chris.Lord@icr.ac.uk).

ATR (Ataxia-Telangiectasia Mutated (ATM) and Rad3-related protein kinase), is a critical component of the cellular DNA damage response (DDR)[1]. ATR is activated by regions of single-stranded DNA, some of which occur as a result of replication stress[2–4]. Oncogene activation can induce replication stress and a reliance upon an ATR checkpoint function; this provides one rationale for the use of small molecule ATR inhibitors (ATRi) as cancer therapeutics[5]. Potent and specific ATRi have been discovered including EPT-46464 (ref. 6), AZ20 (AstraZeneca)[7], VE-821 and VX-970 (VE-822) (Vertex), some of which are currently in Phase I clinical trials[5]. In pre-clinical studies, VE-821 enhances the cytotoxic effects of a number of DNA damaging agents in tumour cells that have defects in the ATM/p53 pathway[8–11], suggesting that ATRi might have clinical utility as chemo-sensitizing agents. However, in what context ATRi might be used as single agents is less clear. Previous studies have demonstrated that alterations in canonical DDR/cell cycle checkpoint genes (ERCC1 (ref. 12), XRCC1 (ref. 13), CDC25A[14] and ATM[15,16]) have the potential to act as predictive biomarkers of single-agent ATRi sensitivity. However, it is not yet clear whether processes beyond canonical DDR, and in particular loss of tumour suppressor genes, might also predict for ATRi sensitivity. Therefore, as ATRi enter Phase 1 clinical trials, it is clear that there is a pressing need to identify clinically useful biomarkers of sensitivity[5].

The SWI/SNF chromatin-remodelling complex is composed of multiple components, including proteins such as ARID1A, ARID1B, SMARCA4 and SMARCB1 that have tumour suppressor roles[17]. These complexes utilize ATP to modify chromatin architecture by sliding or ejecting nucleosomes from DNA[18]. This activity appears to modulate a number of DNA processes including replication, transcription and DNA repair[19,20]. Two primary versions of SWI/SNF have been isolated from cells, BAF (SWI/SNF-A) and PBAF (SWI/SNF-B)[21], distinguished in part by the DNA-binding component of the complex. BAF complexes interact with DNA via either ARID1A or ARID1B components, while the PBAF complex binds DNA through ARID2 (ref. 22). When taken as a group, SWI/SNF components are estimated to be mutated in nearly 20% of all human tumours, making loss of this complex one of the most common alterations in cancer[17].

In this study, we aimed to discover clinically actionable determinants of single-agent ATRi sensitivity. Using large-scale genetic screens we identified the BAF component ARID1A as a synthetic lethal partner of ATR inhibition. We validated the synthetic lethal interaction between ATR and ARID1A using both in vitro and in vivo models. Mechanistically, we found that ATR inhibition exploits a pre-existing DNA decatenation defect in ARID1A mutant tumour cells and causes premature mitotic progression. This leads to large-scale genomic instability and cell death. On the basis of this data, we propose that ARID1A should be assessed as a biomarker of ATRi sensitivity in clinical trials.

## Results

**RNAi screens identify ARID1A as ATRi synthetic lethal partner.** To uncover clinically actionable genetic determinants of single-agent ATRi response, we performed a series of high-throughput RNAi chemosensitization screens where cells were transfected with a library of SMARTPool short interfering (si)RNAs and then exposed to the highly potent and selective ATR catalytic inhibitor VE-821 (Fig. 1a; $K_i = 13$ nM (ref. 23)). For screening we selected the p53 mutant, triple negative (ERα negative, PR negative and ERBB2 negative) breast tumour cell line HCC1143, based on previous work suggesting that ATRi might have utility in TP53 mutant cancers[6,9,24,25]. To model the effect of ATRi on normal cells, we also screened the non-tumour, mammary epithelial cell

model, MCF12A. We confirmed that both cell lines retained a functional ATR activation pathway by assessing cisplatin-induced ATR p.T1989 autophosphorylation[26,27] (Supplementary Fig. 1A,B). To identify clinically actionable effects, the RNAi library we used encompassed 1,280 siRNA SMARTPools (four siRNAs per gene in each pool) targeting either recurrently mutated genes in cancer[28], kinases, due to their inherent tractability as drug targets, and DDR genes[29], given the potential for ATRi to enhance defects in these processes[6,9] (Supplementary Data 1). HCC1143 and MCF12A cells were transfected in a 384-well plate format using the siRNA library. Cells were then exposed to a sub-lethal concentration of VE-821 (1 µM, Supplementary Fig. 1C) or vehicle (DMSO) for a subsequent 4 days, at which point cell viability was estimated using CellTitre-Glo Reagent (Promega; Fig. 1a).

We used data from replica ($n = 3$) screens to calculate drug effect (DE) Z-scores (see Methods) describing the effect of each siRNA SMARTPool on VE-821 sensitization (Fig. 1b)[30]. This approach identified 150 significant (DE $Z < -2$) VE-821 sensitivity-causing effects in HCC1143 cells and 179 in MCF12A cells (Fig. 1b,c and Supplementary Data 2). Consistent with previous observations[9,12,15,16], we observed significant VE-821 sensitization associated with siRNA SMARTPools designed to target either ATR, the ATR activation factors RAD17, RAD1 and RAD9A, ERCC4 or ATM (Supplementary Fig. 1D,E), giving us confidence in the results from the screens.

To identify ATRi synthetic lethal effects operating in diverse genetic backgrounds, we compared the HCC1143 and MCF12A data and identified 30 siRNA SMARTPools that caused VE-821 sensitivity in both cell lines (Supplementary Data 2). This analysis identified several novel ATR synthetic lethal partner genes involved in DNA damage/repair including those targeting components of the HR/Fanconi Anaemia pathway (FANCE, SLX4, PALB2), DNA mismatch repair (MSH4, PMS2) and trans-lesion synthesis (RAD18) pathways (Fig. 1c and Supplementary Data 2). Amongst the common hits we also identified seven siRNA targeting tumour suppressor genes (ATM, FANCE, GCP3, IDH1, PALB2, PMS2 and ARID1A; Fig. 1c,d). The observation that silencing ARID1A sensitized cells to ATRi was particularly interesting as ARID1A is recurrently mutated in a variety of tumour types (45% ovarian clear cell carcinoma (OCCC), 14–19% gastric, bladder and hepatocellular tumours and 2–3% breast tumours[17]).

We validated the ARID1A/ATRi synthetic lethal interaction using a number of orthogonal approaches. In HCC1143 cells, an siRNA SMARTPool targeting ARID1A-reduced ARID1A protein levels and significantly enhanced sensitivity to VE-821 ($P < 0.0001$, analysis of variance (ANOVA), Fig. 1e,f) as did the four individual ARID1A siRNAs ($P < 0.001$, Student's t-test, Fig. 1g,h), suggesting that this was unlikely to be an off-target siRNA effect. Both pooled and individual ARID1A siRNAs also caused sensitivity to an ATRi currently being assessed in Phase 1 clinical trials trials, VX-970 (Fig. 2a), suggesting that the synthetic lethality was not specific to VE-821 (Supplementary Fig. 2A,B). To assess whether constitutive loss of ARID1A function, as would occur in tumours, also resulted in ATRi sensitivity, we used human colorectal HCT116 isogenic cells containing either wild-type ARID1A ($ARID1A^{+/+}$) or homozygous loss-of-function ARID1A mutations (p.Q456*/p.Q456*; referred to as $ARID1A^{-/-}$; Fig. 2b). In clonogenic survival assays exposure to three distinct ATRi (VE-821, VX-970 and AZ20) all significantly impaired the survival of $ARID1A^{-/-}$ cells compared with $ARID1A^{+/+}$ ($P < 0.0001$, ANOVA, Fig. 2c–f). The $ARID1A^{-/-}$ selective toxicity of ATRi was also observed in short-term (5-day exposure) viability assays using all three inhibitors ($P < 0.001$, ANOVA, Fig. 2g–i). In contrast to our

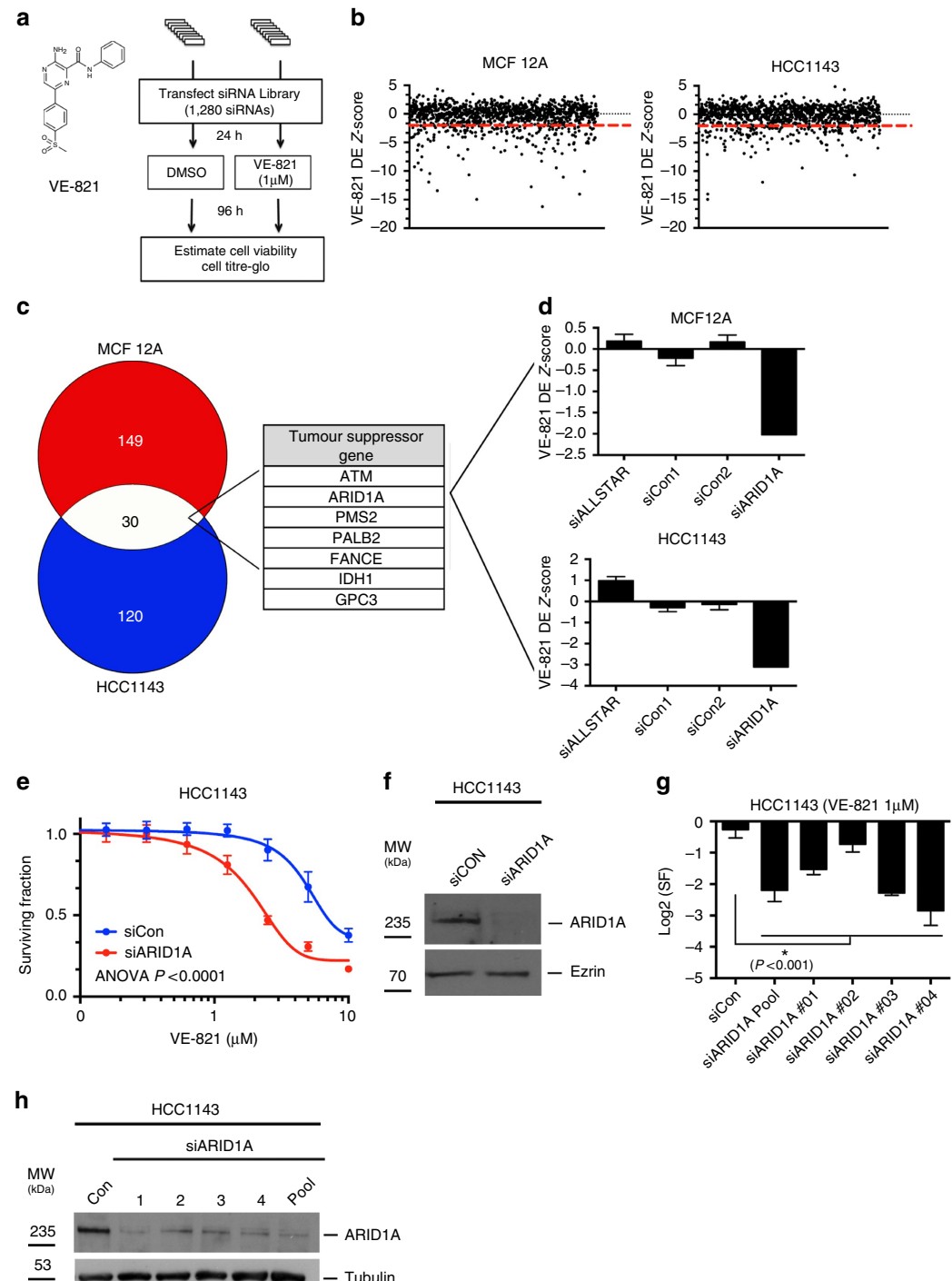

**Figure 1 | RNAi screen reveals genetic determinants of ATRi sensitivity.** (**a**) Structure of VE-821 and schematic representation describing workflow for parallel VE-821 chemosensitization screens in MCF12A and HCC1143 cells. (**b**) Scatter plots of VE-821 Drug Effect (DE) *Z*-scores from all 1280 siRNAs SMARTpools used in the chemosensitization screens. The DE *Z*-score threshold of $-2$ (dotted red line) was used for defining candidate/unvalidated synthetic lethal interactions. (**c**) Venn diagram of DE $< -2$ VE-821 sensitization hits in MCF12A and HCC1143 cells. Numbers shown indicate number of sensitization genes. Amongst the 30 genes with DE *Z*-scores $< -2$ in both cell lines, seven well-established tumour suppressor genes were identified. (**d**) Bar charts illustrating DE *Z*-scores for control, non-targeting, siRNAs (siALLSTAR, siCon1, siCon2) and *ARID1A* SMARTPool siRNAs in the chemosensitization screens. Values shown are medians from triplicate screens. Error bars represent s.d. (**e**) Three-hundred eighty-four-well plate cell survival data from HCC1143 cells transfected with siRNA targeting *ARID1A* (red) or siCon (blue). Twenty four hours after transfection, cells were exposed to VE-821 for 5 continuous days. Error bars represent s.d. ($n = 16$) and results are representative of three biological replicates. Survival curve siARID1A versus siCon *P* value $< 0.0001$, ANOVA. (**f**) Western blot illustrating ARID1A protein silencing from experiment (**e**). (**g**) Bar chart illustrating the Log2 surviving fractions (Log2(SF)) of HCC1143 cells transfected with the indicated individual siRNAs and exposed to VE-821 (1 μM) for 5 days. Error bars represent s.d. and *P* values of $< 0.001$, Student's *t*-test, for each siRNA compared with siCon. (**h**) Western blot illustrating ARID1A protein silencing from experiment (**g**).

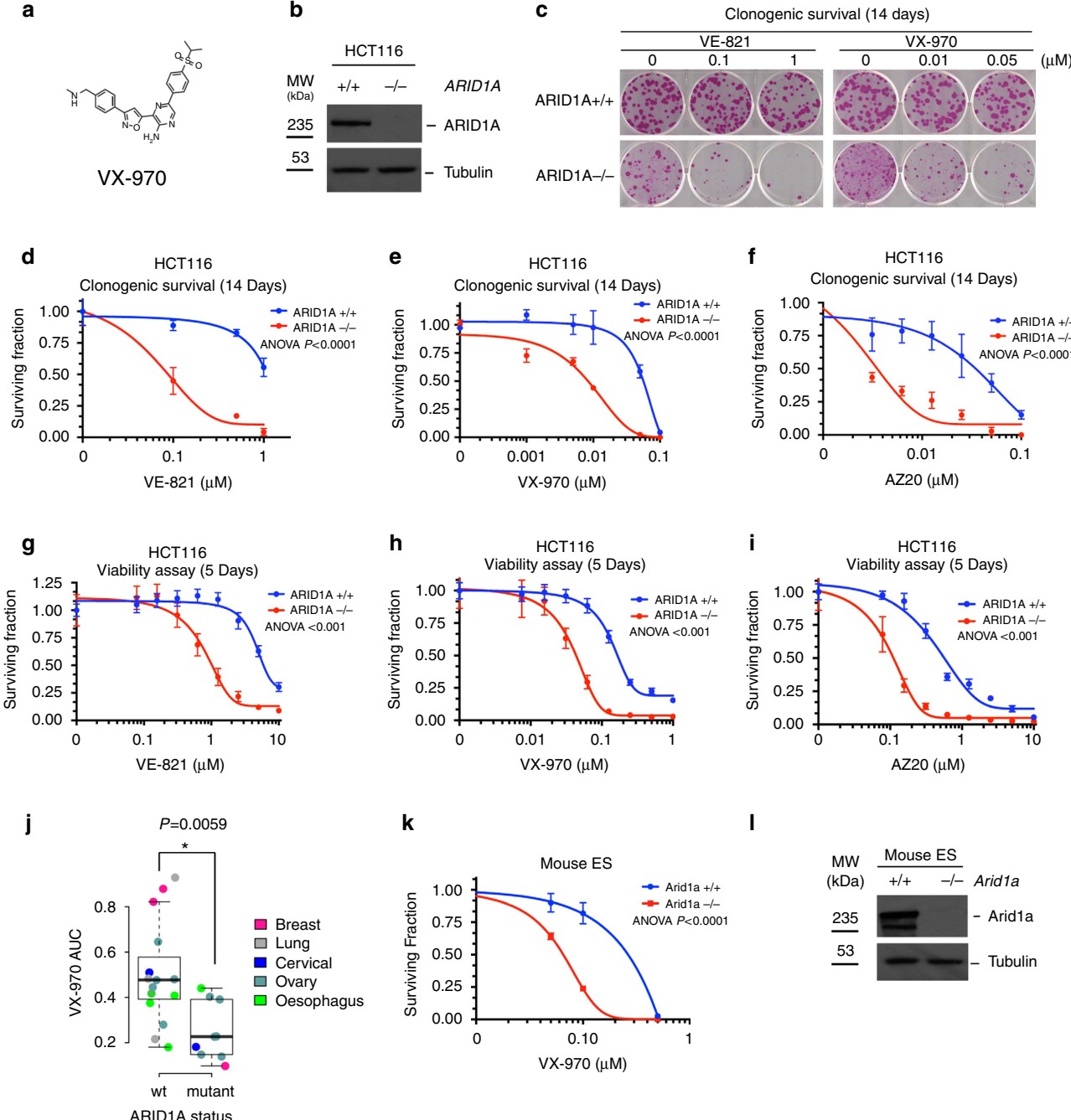

**Figure 2 | *In vitro* ARID1A/ATR synthetic lethality.** (**a**) Chemical structure of VX-970. (**b**) Western blot of ARID1A in human ARID1A isogenic HCT116 cells. (**c**) Image of colonies in six-well-plate clonogenic assay. HCT116 ARID1A isogenic ( $+/+$ and $-/-$ ) cell lines were exposed to increasing concentrations of VE-821 (0, 0.1, 1 μM) or VX-970 (0, 0.01, 0.05 μM) for 14 days. (**d-f**) Dose–response clonogenic survival curves of HCT116 ARID1A isogenic ( $+/+$ and $-/-$ ) cell lines exposed to increasing concentrations of VE-821 (**d**), VX-970 (**e**) and AZ-20 (**f**) for 14 days. Error bars represent s.d. ( $n=3$ ), ANOVA $P$ value of $<0.0001$, results are representative of triplicate biological experiments. (**g-i**) Dose–response survival curves of HCT116 ARID1A isogenic ( $+/+$ and $-/-$ ) cell lines exposed to increasing concentrations of VE-821 (**g**), VX-970 (**H**) or AZ-20 (**I**) for 5 days. Cell viability was estimated by CellTitre-Glo reagent. Error bars represent s.d. ( $n=16$ ), ANOVA $P$ value of $<0.001$, results are representative of triplicate biological experiments. (**j**) Area under curve (AUC) box whisker comparison plot for human tumour cell lines exposed to VX-970 for 5 days. *ARID1A* wild-type tumour cell lines ( $n=15$ ) were compared with *ARID1A* mutant cell lines ( $n=9$ ). $P=0.00594$ median permutation test. (**k**). Dose–response clonogenic survival curve of mouse *Arid1a* isogenic ES cell lines. Experiment was performed as per (**c**). Error bars represent s.d. ( $n=3$ ), ANOVA $P$ value of $<0.0001$, results are representative of triplicate biological experiments. (**l**). Western blot of Arid1a protein expression in mouse ES *Arid1a* isogenic cells.

observations using ATRi, HCT116 ARID1A isogenic cells had a similar level of sensitivity to other common chemotherapeutic agents such as methotrexate and taxol, suggesting that

$ARID1A^{-/-}$ cells are not globally sensitive to exogenous agents (Supplementary Fig. 2C,D). In addition, we noted that heterozygous HCT116 $ARID1A^{+/-}$ cells expressed an intermediate

level of ARID1A protein compared with homozygous wild-type and homozygous mutant cells and exhibited an intermediate level of ATRi sensitivity (Supplementary Fig. 2E,F). This suggested that an *ARID1A* gene dosage effect might possibly influence ATRi sensitivity.

The ATRi sensitivity in HCT116 *ARID1A* $^{-/-}$ cells not only validated the synthetic lethal interaction between ARID1A and ATR inhibition but also established that the effect was likely not restricted to cells from a breast lineage such as HCC1143 or MCF12A used in the high-throughput screens. To further investigate the generality of this effect we determined ATRi sensitivity in a genetically and histologically diverse panel of tumour cell lines. In this analysis, we found that the presence of loss-of-function *ARID1A* mutations in tumour cell lines was associated with sensitivity to VX-970 ($P = 0.0059$, Student's *t*-test, Fig. 2j and Supplementary Data 3), although we were statistically underpowered to confirm whether or not the synthetic lethality was any more or less profound in particular tumour types in this analysis. To address whether the ATR/ARID1A synthetic lethal interaction was limited to human cell line models, we also studied isogenic *Arid1a* $^{+/+}$ and *Arid1a* $^{-/-}$ mouse embryonic stem (ES) cells[31]. In clonogenic assays both VX-970 and VE-821 selectively targeted *Arid1a* $^{-/-}$ ES cells ($P < 0.0001$, ANOVA, Fig. 2k,l and Supplementary Fig. 2E). We next compared how the observed ATR/ARID1A synthetic lethality compared with other ATRi related synthetic lethalities and other reported ARID1A-selective agents. The magnitude of ATRi sensitivity associated with *ARID1A* mutant cells was comparable, if not greater than that seen with other candidate biomarkers of ATRi sensitivity such as *ATM* (Supplementary Fig. 2F). Furthermore, we found that the *ARID1A* $^{-/-}$ selectivity of VX-970 was also more profound than other proposed ARID1A-targeted agents such as PARP inhibitors (PARPi) or cisplatin[32,33] (Supplementary Fig. 3).

We originally identified the ATR/ARID1A synthetic lethal effect in p53 mutant HCC1143 and p53 wild-type MCF12A cells and confirmed this synthetic lethality in p53 wild-type HCT116 and mouse ES cells (Figs 1 and 2). This suggested that the synthetic lethal effect was somewhat independent of p53 status. Silencing of *ARID1A* using siRNA significantly sensitized HCT116 *TP53* $^{-/-}$ cells to VX-970 ($P < 0.001$, ANOVA, Supplementary Fig. 4A), suggesting that the absence of p53 function did not negate the ATR/ARID1A synthetic lethality. In addition, silencing of *TP53* using siRNA in p53 wild-type HCT116 *ARID1A* $^{-/-}$ cells also did not alter the sensitivity to VX-970 (Supplementary Fig. 4B). Collectively this data suggested that p53 might not be a key modulator of the ATR/ARID1A synthetic lethality.

To address the possibility that the ARID1A/ATRi synthetic lethality might be specific to catalytic inhibition of ATR, we exploited data describing the sensitivity of 86 human tumour cell lines to SMARTPool siRNAs targeting 720 kinase-coding genes[34,35]. We found that the presence of loss-of-function *ARID1A* mutations in tumour cell lines was associated with greater sensitivity to *ATR* siRNA ($P = 0.008$, Median Permutation test, Supplementary Fig. 4C). To confirm this, we silenced *ATR* using siRNA in HCT116 *ARID1A* $^{-/-}$ and *ARID1A* $^{+/+}$ cells, and found that *ATR* siRNA selectively targeted *ARID1A* $^{-/-}$ cells ($P < 0.05$, Student's *t*-test, Supplementary Fig. 4D,E). Collectively, this data suggested that inhibition of ATR function, either by small molecule inhibition or by gene silencing was synthetically lethal with *ARID1A* deficiency.

***In vivo* ATR/ARID1A synthetic lethality.** We next assessed whether the clinical ATRi, VX-970, could inhibit ARID1A-deficient tumours *in vivo*. To do this, we generated cohorts

of mice with established xenograft tumours derived from either HCT116 *ARID1A* $^{+/+}$ or *ARID1A* $^{-/-}$ cells. Once tumours had established, mice were treated with either VX-970 or drug vehicle for 40 days (Fig. 3a). We found that, compared with vehicle treatment, VX-970 had no effect on *ARID1A* $^{+/+}$ tumours ($P = 0.45$, ANOVA, Fig. 3b) but significantly inhibited the growth of *ARID1A* $^{-/-}$ tumours ($P = 0.024$, ANOVA, Fig. 3b). VX-970 treatment also had a greater impact on *ARID1A* $^{-/-}$ tumours compared with *ARID1A* $^{+/+}$ tumours ($P = 0.006$, ANOVA, Fig. 3b). In a subsequent experiment, we assessed whether VX-970 could impair the establishment of tumours by treating mice with VX-970 immediately after xenografting HCT116 cells, as opposed to waiting for tumours to establish before initiating treatment (Fig. 3c). We found that VX-970 prevented the establishment of HCT116 *ARID1A* $^{-/-}$ xenografts (frequency of *ARID1A* $^{-/-}$ tumour formation = 27% for VX-970-treated mice versus 73% in vehicle-treated mice, $P = 0.027$, Fisher's exact test), but had no impact on the establishment of *ARID1A* $^{+/+}$ xeno-grafts (frequency of *ARID1A* $^{-/-}$ tumour formation = 80% for VX-970-treated mice versus 87% in vehicle-treated mice, $P = 1$ ns by Fisher's exact test, Fig. 3c,d). In this experiment, early initiation of VX-970 treatment also dramatically slowed the growth of *ARID1A* $^{-/-}$ tumours ($P = 0.015$, ANOVA), but did not impair *ARID1A* $^{+/+}$ xenografts ($P = 0.63$, ANOVA, Fig. 3e,f). We also determined the *in vivo* efficacy of VX-970 in mice with established tumours derived from a tumour cell line with naturally occurring *ARID1A* mutations (TOV21G, *ARID1A* p.548fs/p.756fs). In *in vitro* studies, we found that the TOV21G human OCCC cell line was more sensitive to both VE-821 and VX-970 than the *ARID1A* wild-type OCCC cell line RMG1 (Fig. 3g, $P < 0.0001$, ANOVA and Supplementary Fig. 4F) and also exhibited a DNA damage (γH2AX; Supplementary Fig. 4G) and apoptotic response to ATRi exposure (Fig. 3h,i). *In vivo*, VX-970 treatment significantly inhibited the growth of established TOV21G tumours compared with vehicle administration ($P = 0.014$, ANOVA, Fig. 3j–l). Although the scale of anti-tumour effect in TOV21G xenografts was not as profound as in HCT116 *ARID1A* $^{-/-}$ xenografts, the effect of ATRi in this setting did suggest that the ARID1A/ATR synthetic lethality could be exploited *in vivo* and warrants further investigation.

**ARID1A loss results in TOP2A and cell cycle defects.** In the course of establishing what differences between HCT116 *ARID1A* $^{+/+}$ and *ARID1A* $^{-/-}$ cells could explain the selective toxicity of ATR inhibition, we found that under logarithmic growth conditions, and in the absence of ATRi, the proportion of $G_2/M$ cells was higher in *ARID1A* $^{-/-}$ cells compared with *ARID1A* $^{+/+}$ cells ($P = 0.033$, Student's test, Fig. 4a). To investigate this further, we synchronized cells at the $G_1$/early S phase boundary using a double thymidine block and then followed cell cycle progression once the thymidine block was removed (Fig. 4b). We found that, compared with *ARID1A* $^{+/+}$ cells, *ARID1A* $^{-/-}$ cells had a delayed progression through S phase and a slower progression from $G_2$ into mitosis (8–10 h after block—Fig. 4b). In asynchronous cell cultures, we found that *ARID1A* $^{-/-}$ cells exhibited lower levels of phosphorylated Histone H3 (Ser-10), a marker of mitotic entry, compared with *ARID1A* $^{+/+}$ cells ($P = 0.035$, Student's *t*-test, Fig. 4c). Silencing of *ARID1A* by siRNA SMARTpool in *ARID1A* $^{+/+}$ cells also caused a significant decrease in Histone H3 phosphorylation ($P = 0.00015$, Student's *t*-test, Fig. 4c). We did however note that the extent of phospho-H3 reduction caused by ARID1A inhibition was more extensive in ARID1A siRNA SMARTpool trans-fected cells than in *ARID1A* $^{-/-}$ cells (Fig. 4c). It is possible that this could be due to a modest difference in the pheno-

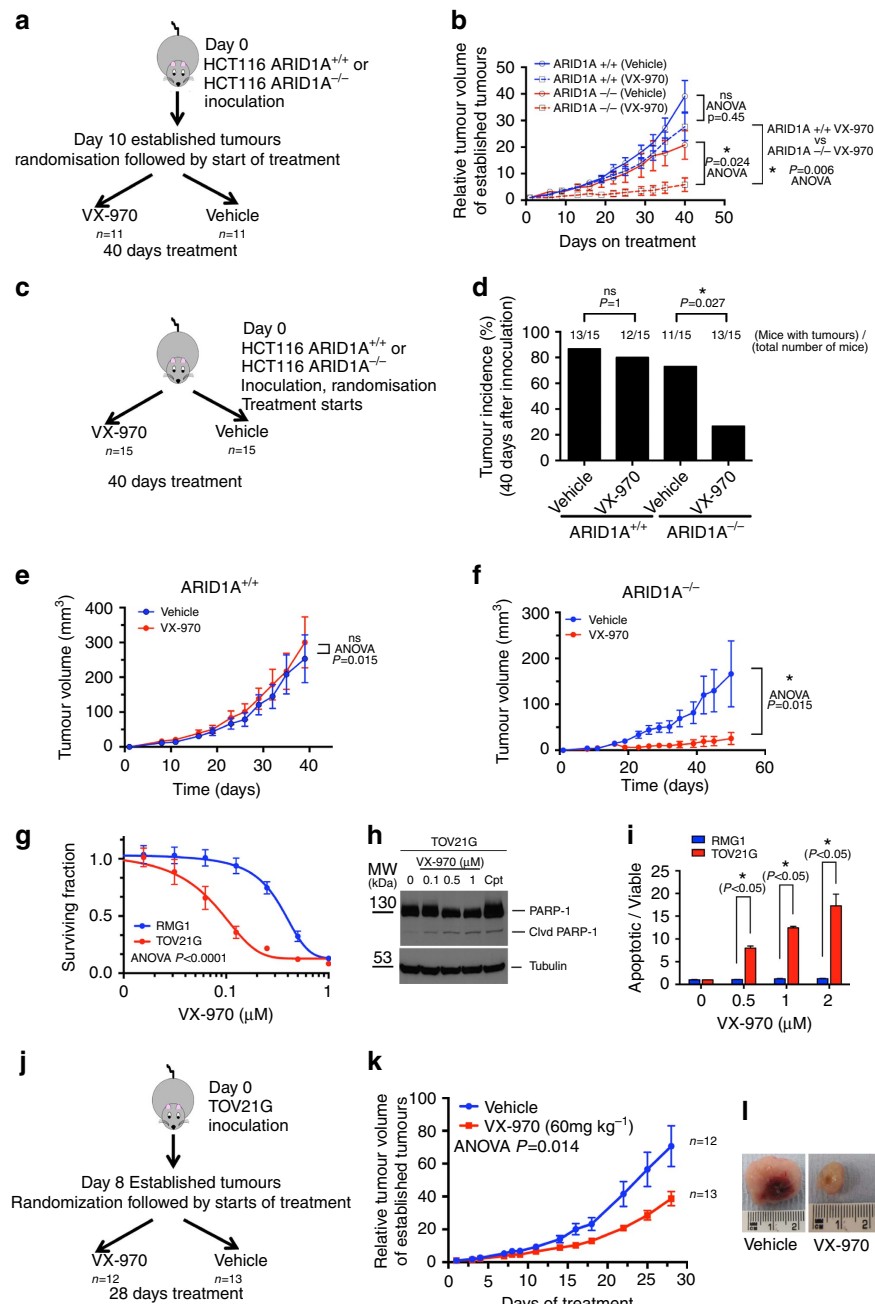

**Figure 3 | In vivo ARID1A/ATR synthetic lethality. (a).** Schematic representation of VX-970 therapy experiment in mice bearing established HCT116 *ARID1A*$^{+/+}$ and *ARID1A*$^{-/-}$ xenografts. Mice were then randomized to treatment cohorts of either VX-970 (60 mg kg$^{-1}$, 4 × weekly by oral gavage) or vehicle treatments. N = 11 mice in each cohort. Mice were treated for a subsequent 40 days. Tumour volume was monitored thrice weekly. **(b)** Relative tumour volume plot from experiment (A) showing efficacy of VX-970 and selectivity for *ARID1A*$^{-/-}$ xenografts * P = 0.024, ANOVA, and a greater efficacy of VX-970 on *ARID1A*$^{-/-}$ xenografts compared with *ARID1A*$^{+/+}$ xenografts *P = 0.006, ANOVA. **(c)** Schematic representation of tumour incidence experiment in mice with non-established HCT116 *ARID1A*$^{+/+}$ and *ARID1A*$^{-/-}$ xenografts. N = 15 mice per cohort. Tumour volume was monitored thrice weekly and overall tumour incidence was assessed 40 days later. **(d)** Bar chart of tumour incidence 40 days after implantation of HCT116 *ARID1A*$^{+/+}$ or *ARID1A*$^{-/-}$ cells. Numbers above bars indicate the proportion of animals in which a detectable xenograft formed. * P = 0.027 Fisher's exact test. **(e,f)** Tumour growth from all mice in experiment described in **c**. * P = 0.015 ANOVA. **(g).** VX-970 dose–response curves from TOV21G (*ARID1A* mutant) and RMG1 (*ARID1A* wild-type) tumour cells. Cells were exposed to VX-970 for 5 days. ANOVA P value of < 0.0001. **(h)** Western blot illustrating PARP-1 cleavage in TOV21G cells exposed to increasing concentrations of VX-970 for 24 h before cell lysis. As a positive control cells were exposed to camptothecin (Cpt; 1 μM, 24 h) before cell lysis. PARP-1, cleaved (Clvd) PARP-1 (85 kDa fragment) and tubulin were detected by western blot. **(i)** Bar chart illustrating apoptotic fraction in RMG1 and TOV21G cells exposed to increasing concentrations of VX-970 for 24 h. *Student's *t*-test, P < 0.05. **(j)** Schematic representation of VX-970 therapy experiment in mice bearing established TOV21G. Experiment performed as per **(a)** but with only 28 days treatment. **(k)** Relative tumour volume plot from **(j)** showing efficacy of VX-970 in TOV21G xenografts *P = 0.014, ANOVA. **(l).** Images of vehicle and VX-970-treated TOV21G xenografts after 28 days treatment.

type between acute loss of ARID1A expression (via siRNA) as opposed to long-term ARID1A deficiency as is the case in the $ARID1A^{-/-}$ cells. We also noted increased levels of cytoplasmic cyclin B1 in $ARID1A^{-/-}$ cells (Supplementary Fig. 4H) a phenotype associated with initiation of a $G_2/M$ cell cycle checkpoint[36]. Collectively, this data suggested that loss of ARID1A

caused a reduced rate of S phase progression and an increased utilization of a $G_2/M$ cell cycle checkpoint.

Mouse ES cells with an inducible BAF complex defect caused by loss of Brg1 (Smarca4), have delayed progression through a $G_2/M$ cell cycle checkpoint, a defect in the ability to separate DNA sister chromatids after DNA replication (DNA

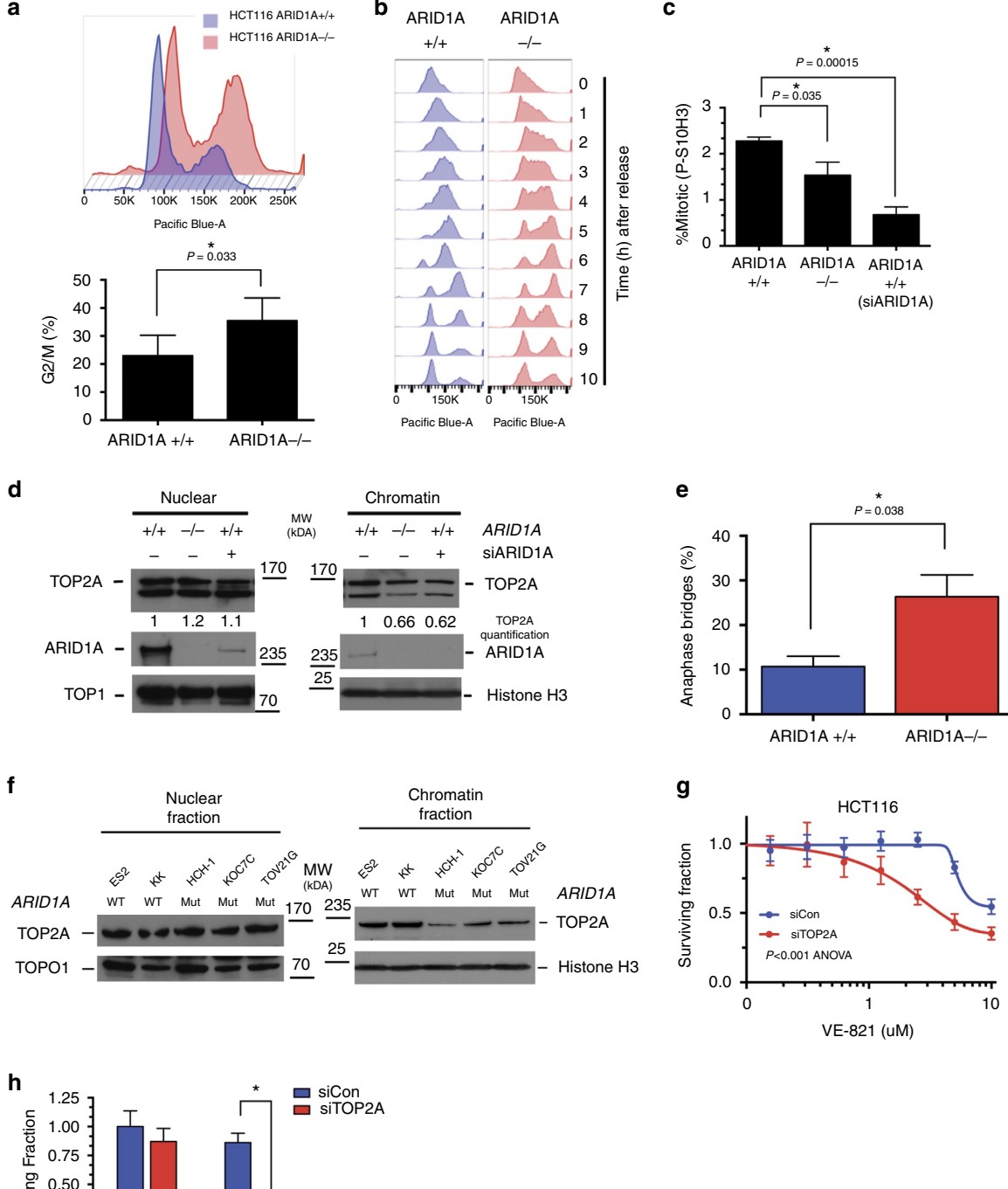

decatenation), and reduced localization of the topoisomerase enzyme, TOP2A, to DNA[37]. TOP2A modulates the topological structure of DNA and is critical to processes such as chromosome condensation, chromatid separation, DNA transcription and replication (reviewed in refs 38,39). We found that HCT116 $ARID1A^{-/-}$ cells exhibited lower levels of chromatin-bound TOP2A, compared with isogenic $ARID1A^{+/+}$ cells (Fig. 4d). In addition RNAi-mediated silencing of $ARID1A$ in $ARID1A^{+/+}$ cells also reduced chromatin-bound TOP2A (Fig. 4d). We also found that the TOP2A defect in $ARID1A$ defective HCT116 cells was associated with an increased frequency of anaphase bridges ($P = 0.038$, Student's $t$-test, Fig. 4e), a biomarker of defective DNA decatenation, a process controlled by TOP2A[37,40]. The TOP2A defect was also apparent amongst a panel of OCCC cell lines, where those with $ARID1A$ mutations exhibited a reduction in chromatin-bound TOP2A, despite expressing similar amounts of total TOP2A (Fig. 4f). We also found that siRNA silencing of $TOP2A$ using SMARTPool siRNAs in $ARID1A^{+/+}$ HCT116 cells caused sensitivity to both VE-821 and VX-970 ($P < 0.001$, ANOVA, Fig. 4g and Supplementary Fig. 4I,J). Silencing of TOP2A by SMARTPool siRNA in HCT116 $ARID1A^{-/-}$ cells led to high levels of cell death, presumably because of the pre-existing TOP2A defect in these cells (Fig. 4h).

It seemed possible that the defects in TOP2A localization and cell cycle progression in $ARID1A$ mutant cells would generate an enhanced necessity for ATR function that could be therapeutically exploited with ATRi. TOP2A alleviates many of the topological problems caused by chromosomal metabolism such as catenated DNA and DNA transcription/replication interference[41]. It seemed possible that cell division in the presence of a TOP2A defect and a failure to adequately resolve such topological problems could cause genomic instability and ultimately impair the fitness of cells. In HCT116 cell cultures synchronized at the late $G_1$/early S phase boundary using a double thymidine block, we found that exposure to VX-970 accelerated the rate of $G_2$ exit, so that $ARID1A^{-/-}$ cells no longer exhibited the $G_2$/M cell cycle progression delay seen in the absence of ATRi (Fig. 5a). In addition, VX-970 exposure also resulted in reduced cytoplasmic Cyclin B1 in both $ARID1A^{+/+}$ and $ARID1A^{-/-}$ cells (Supplementary Fig. 4H). Despite both wild type and ARID1A null cells displaying increased mitotic entry in response to VX-970, ATRi exposure led to increased anaphase bridges in $ARID1A^{-/-}$ cells compared with $ARID1A^{+/+}$ ($P = 0.023$, Student's $t$-test, Fig. 5b) an effect also observed in ARID1A

mutant TOV21G cells ($P = 0.006$ by Student's $t$-test, Fig. 5c). VX-970 exposure also led to a significant enhancement in the frequency of chromosomal aberrations in $ARID1A^{-/-}$ cells compared with wild-type cells ($P = 0.0007$, Student's $t$-test, Fig. 5d,e). In addition, we found that the VX-970-induced γH2AX response, a marker of DNA damage, was more pronounced in $ARID1A^{-/-}$ cells (Fig. 5f). A similar γH2AX response was also observed in ARID1A mutant TOV21G cells (Supplementary Fig. 4G). In $ARID1A^{-/-}$ cells exposed to ATRi for a short period (2 h), we found an increase in H2AX phosphorylation in Cyclin A positive as well as negative cells (Supplementary Fig. 5A–D). The presence of VX-970-induced H2AX phosphorylation in S phase cells was confirmed by western blotting in synchronized cells as they transitioned through S phase (Supplementary Fig. 5E). Taken together, this suggested that the γH2AX response to ATRi in $ARID1A^{-/-}$ cells occurred in S/$G_2$ phases (Cyclin A positive) as well as in $G_1$ phase (Cyclin A negative) of the cell cycle, implying that processes occurring in multiple phases of the cell cycle could contribute to the synthetic lethal phenotype observed. It is possible that the γH2AX response in Cyclin A negative cells could be initiated in $G_1$ itself or could be the result of DNA damage occurring during mitosis becoming apparent as cells transit into $G_1$ during the 2-h ATRi exposure. We also noted that VX-970 exposure caused an apoptotic response in $ARID1A$-deficient cells, as shown by higher caspase-3/7 activation in $ARID1A^{-/-}$ cells compared with $ARID1A^{+/+}$ cells ($P < 0.05$, Student's $t$-test, Fig. 5g) and in TOV21G compared with RMG1 cells (Fig. 3i). We also assessed the induction of apoptosis in ATRi exposed cells by measuring PARP-1 cleavage. Camptothecin elicited similar levels of cleavage in both $ARID1A^{-/-}$ and $ARID1A^{+/+}$ cells, suggesting both genotypes possessed a functional apoptotic response. In contrast, VX-970 caused far higher levels of PARP-1 cleavage in $ARID1A^{-/-}$ cells than in $ARID1A^{+/+}$ cells (Fig. 5h). From this data we concluded that loss of ARID1A function results in: (i) a defect in the ability of cells to recruit TOP2A to chromatin; and (ii) cell cycle progression defects in both S and $G_2$/M phases of the cell cycle. It seems possible that these factors combined or in isolation might render tumour cells sensitive to small molecule ATRi as these agents impair the ability of cells to mount adequate DDRs, while at the same time accelerating mitotic entry (Fig. 5i).

In addition to the model proposed (Fig. 5i), we also assessed whether other mechanisms of ATRi sensitivity might explain, or at least contribute, to the ARID1A synthetic lethality. The BAF

**Figure 4 | ARID1A-deficient cells depend on a G2/M checkpoint due to a chromosomal decatenation defect.** (**a**) Histogram of the cellular DAPI-stained DNA content, determined by FACS, in asynchronous HCT116 $ARID1A^{+/+}$ (blue) and $ARID1A^{-/-}$ cells (red). Bar chart illustrating the percentage of cells in $G_2$/M phase of asynchronously growing HCT116 $ARID1A^{+/+}$ and $ARID1A^{-/-}$ cells. Asterisk indicates statistical significance ($P = 0.033$, Student's $t$-test), error bars represent s.d., from four independent experiments. (**b**) Histogram of the cellular DAPI-stained DNA content, determined by FACS, in HCT116 $ARID1A^{+/+}$ and $ARID1A^{-/-}$ cells at the indicated time points following release from double thymidine synchronization in $G_1$/early S phase. (**c**) Bar chart illustrating the percentage of mitotic cells with phosphorylation of Histone H3 on Serine 10. Where shown, HCT116 $ARID1A^{+/+}$ cells were transfected with siARID1A. 48 h later cells were fixed and stained with FITC-P-S10 Histone H3 and PI. Mitotic cells (P-S10 positive and 2 N) were quantified by FACS. Asterisk indicates a statistically significant difference by Student's $t$-test between the indicated comparisons. (**d**) Western blot of nuclear and chromatin-bound TOP2A. The indicated cell lines were transfected with siRNA targeting $ARID1A$ or control siRNA. Forty-eight hous later, subcellular fractions were isolated and resultant western blots immunoblotted for the indicated proteins. Total amount of TOP2A was quantified and normalized to $ARID1A^{+/+}$ control siRNA, in each fraction. (**e**) Bar chart illustrating elevated level of anaphase bridges in HCT116 $ARID1A^{-/-}$ cells compared with $ARID1A^{+/+}$. A minimum of 50 anaphases were scored in three biological replicate experiments. * $P = 0.038$, Student's $t$-test. (**f**) Western blot of nuclear and chromatin-bound TOP2A from OCCC cell lines, with the indicated $ARID1A$ status. Fractionation was performed as per (**d**). (**g**) Three hundred eighty four-well plate cell survival data from HCT116 cells transfected with siRNA targeting $TOP2A$ (red) or siCon (blue). Twenty four hours after transfection, cells were exposed to VE-821 for 5 continuous days. Error bars represent s.d. Survival curves siTOP2A versus siCon $P$ value $< 0.001$, ANOVA. (**h**). Three hundred eighty four-well plate cell survival data from HCT116 $ARID1A^{+/+}$ and HCT116 $ARID1A^{-/-}$ cells transfected with siRNA targeting $TOP2A$ (red) or siCon (blue). Viability was estimated 5 days after transfection. Error bars represent s.d. and asterisks indicate a statistically significant difference ($P < 0.001$) by Student's $t$-test between the indicated comparisons. FACS, fluorescence-activated cell sorting.

chromatin-remodelling complex plays a central role in the control of gene transcription[20,42]. We therefore also assessed whether loss of *ARID1A* caused reduced transcription of any previously established ATR synthetic lethal genes that could also explain the ATR/ARID1A synthetic lethal effect. By comparing trans-

criptomic profiles of HCT116 $ARID1A^{+/+}$ and $ARID1A^{-/-}$ cells (see Methods) we detected a statistically significant reduction in *ARID1A* mRNA expression in $ARID1A^{-/-}$ cells, presumably due to nonsense mediated decay of the mutant transcript (Supplementary Data 4). We did not however, find significant

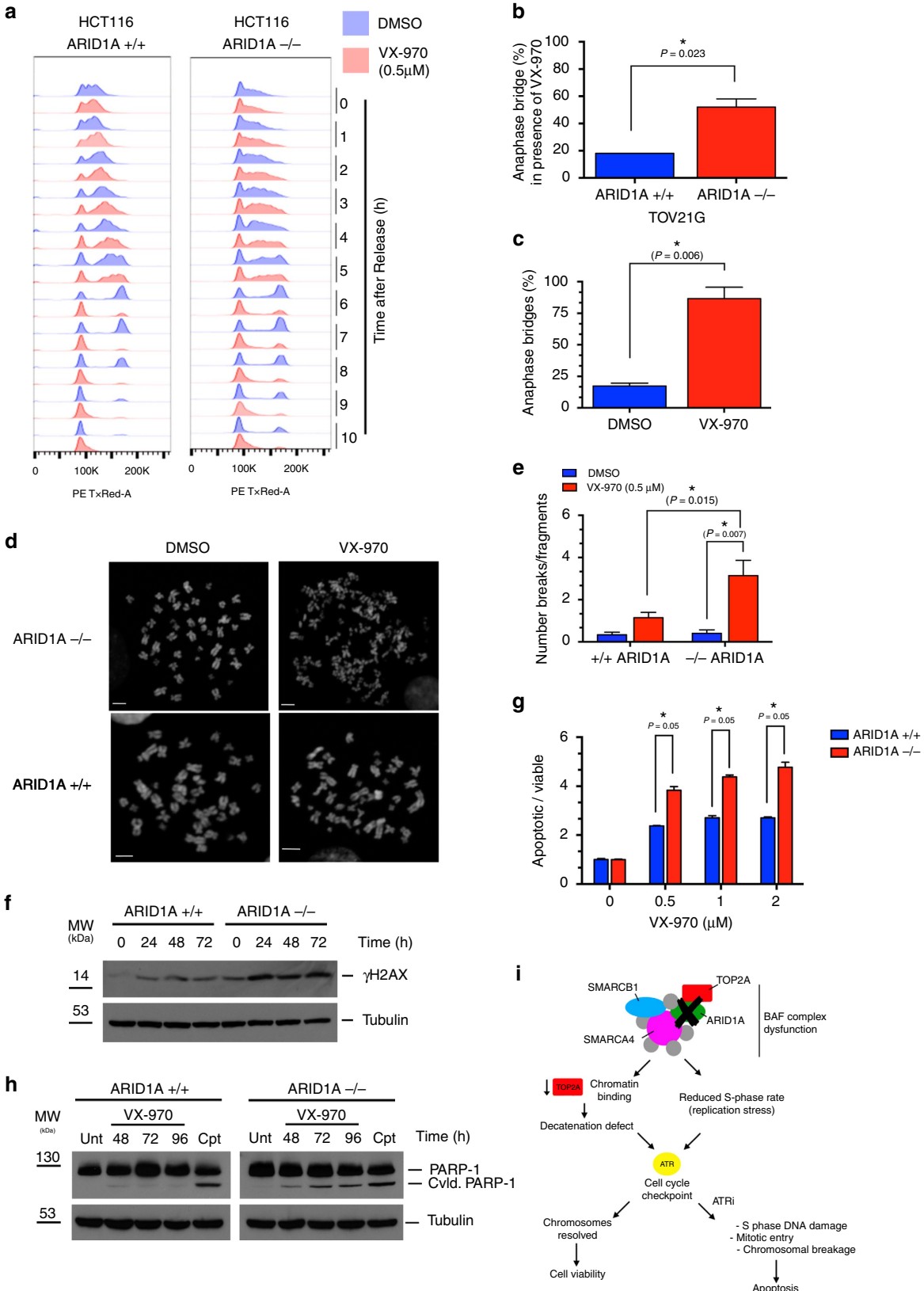

reduction in mRNA levels of *ATM*, *XRCC1*, *ATR* or the ATR activating genes *RAD17*, *HUS1* or *RAD9A* in the *ARID1A*$^{-/-}$ cells when compared with the HCT116 *ARID1A*$^{+/+}$ isogenic model (Supplementary Data 4). *ERCC1* defects have previously been associated with ATRi sensitivity[12] but in this analysis we found *ERCC1* transcript levels to be elevated in the *ARID1A* defective cells, not decreased (Supplementary Data 4). Although we cannot formally exclude the possibility that a transcriptional mechanism contributes to the ARID1A/ATR synthetic lethality, we were unable to identify significant changes in gene expression in isogenic models that might explain this.

We also assessed whether defects in ATR activity in ARID1A defective cells might explain the synthetic lethal effects observed. Shen *et al.* recently demonstrated that ATR and ARID1A directly interact and loss of this interaction in HCT116 *ARID1A*$^{-/-}$ tumour cells results in impaired ATR activation (as measured by ATR autophoshorylation) in response to ionizing radiation, although ATR signalling in response to agents that stall replication forks such as hydroxyurea (HU) and ultraviolet radiation appeared normal[32]. We assessed whether ATR activity was impaired in HCT116 *ARID1A*$^{-/-}$ tumour cells as well as in a panel of human tumour cell lines characterized according to their *ARID1A* status (Supplementary Fig. 1A,B). We found that in response to cisplatin, a platinum salt drug that causes replication fork stress[43], HCT116 *ARID1A*$^{-/-}$ and tumour cell lines with endogenous *ARID1A* mutations exhibited similar protein expression of ATR to ARID1A wild-type cells and clear ATR p.T1989 autophosphorylation (Supplementary Fig. 1A,B). This data suggested that while ATR responses in HCT116 *ARID1A*$^{-/-}$ tumour cells to IR are impaired[32], this was not the case in a variety of tumour cells exposed to cisplatin (Supplementary Fig. 1A,B) or other replication fork stalling lesions[32]. As such, we were unable to explain the ATRi sensitivity of ARID1A defective cells by reason of reduced ATR activity.

## Discussion

Our data suggest that ATRi could have potential as single-agent treatments for ARID1A defective cancers. The highly recurrent nature of *ARID1A* mutations in human cancer and the availability of clinical ATRi suggests that once Phase I clinical trials are complete, biomarker driven proof-of-concept trials could be instigated to assess this hypothesis. These trials could be conducted in cancer types where there is a high frequency of *ARID1A* mutations, such as OCCC, where standard of care therapeutic responses are limited and where few targeted approaches exist.

In Fig. 5i, we present a working model to explain the ARID1A/ATR synthetic lethality. There might of course be additional mechanisms that also contribute to this synthetic lethality, as highlighted earlier. Although we cannot find evidence for ATR activation defects in ARID1A defective cells, we cannot formally exclude the possibility that these processes play a part in the synthetic lethal effect and act in parallel to the TOP2A and cell cycle defects. In addition, cells utilizing the ALT-pathway of telomere maintenance have also been shown to have increased ATRi sensitivity[44], although whether this effect operates in all tumour cells with an ALT defect is not clear[45]. To our knowledge, none of the models used here are ALT-positive. In addition, in Fig. 4, we show that ARID1A defective cells have a delayed progression through the cell cycle and specifically delayed progression through S and G$_2$/M. Cell cycle defects appear to be an inherent characteristic of ARID1A/BAF defective cells, as originally noted by others[31,37]; as such, it seems difficult to eliminate the possibility that some other characteristic(s) associated with reduced proliferation might also influence the sensitivity of ARID1A mutant cells to ATRi. Topoisomerase function has also been implicated in the prevention of deleterious collisions between the transcriptional and replication machineries[41], potentially providing another mechanism that, if defective, could lead to a dependency upon ATR, and could explain the S phase delay in *ARID1A*$^{-/-}$ cells. Finally it is possible that loss of ARID1A, by causing alterations in chromatin structure that lead to an altered transcriptional programme, could impair the expression of other genes critical for cells to survive in the face of ATR inhibition. In this study we compared the transcriptional profiles of HCT116 ARID1A$^{-/-}$ and ARID1A$^{+/+}$ cells but observed no clear defects in the transcript expression of genes known to be associated with ATRi sensitivity (Supplementary Data 4).

Our results, and previous work[37] have detailed a role for ARID1A in chromosomal decatenation through the appropriate localization of TOP2A to chromatin. However, despite this defect in TOP2A, tumour subtypes characterized by a high frequency of *ARID1A* mutations, such as OCCC, have been reported to have relatively 'flat' genomes (that is, relatively few large-scale genomic alterations/rearrangements)[46] compared with other gynaecological malignancies such as high-grade serous ovarian cancers. While the TOP2A defect and 'flat' genomes might possibly be viewed as inconsistent with one another, it is possible that the TOP2A defect in ARID1A mutant tumours has relatively moderate effects (compared with a profound homologous recombination defect for example) on the structure of the genome as it is normally kept in check by proteins such as ATR. Indeed, when

**Figure 5 | ATRi effect on cell cycle progression, chromosomal instability, DNA damage and apoptosis.** (**a**). Histogram of the cellular DAPI-stained DNA content, determined by FACS, in HCT116 *ARID1A*$^{+/+}$ and *ARID1A*$^{-/-}$ cells at the indicated time points following release from synchronization in G$_1$/early S phase by double thymidine block. Cells were released from thymidine block into media containing DMSO (blue) or VX-970 (0.5 μM, red). (**b**). Bar chart illustrating increased level of anaphase bridges in HCT116 *ARID1A*$^{-/-}$ and *ARID1A*$^{+/+}$ cells exposed to VX-970 (0.5 μM, 8 h). Cells were stained with DAPI. A minimum of 50 anaphases were scored in three biological replicate experiments. *P = 0.023, Student's *t*-test. (**c**). Bar chart illustrating frequency of anaphase bridges in TOV21G cells exposed to VX-970 (0.5 μM) or DMSO for 8 h before fixation. Experiment performed as for (**b**). P = 0.006 Student's *t*-test. (**d**) Images of mitotic spreads from HCT116 *ARID1A*$^{+/+}$ and *ARID1A*$^{-/-}$ cells following exposure to either DMSO or VX-970 (1 μM). Scale bar, 20 μm. (**e**) Bar chart illustrating extent of chromosomal abnormalities in HCT116 *ARID1A*$^{+/+}$ and *ARID1A*$^{-/-}$ cells exposed to VX-970 (1 μM). *P < 0.05, Student's *t*-test. (**f**) Western blot illustrating γH2AX in HCT116 *ARID1A*$^{+/+}$ and *ARID1A*$^{-/-}$ cells exposed to VX-970 (0.5 μM) for the indicated time before cell lyses. (**g**) Bar chart illustrating apoptotic fraction in cells exposed to increasing concentrations of VX-970 for 24 h. Experiment as per Fig. 3i. (**h**) Western blot illustrating PARP cleavage in HCT116 *ARID1A*$^{+/+}$ and *ARID1A*$^{-/-}$ cells exposed to VX-970. Experiment performed as per Fig. 3h. (**i**) A model for the proposed mechanism driving the sensitivity of ARID1A-deficient cells to ATRi. Loss of ARID1A function results in: (i) a defect in the ability of cells to recruit TOP2A to chromatin; and (ii) cell cycle progression defects in both S and G$_2$/M phases of the cell cycle. These factors combined or in isolation might render tumour cells sensitive to small molecule ATRi as these agents impair the ability of cells to mount adequate DDRs, while at the same time accelerating mitotic entry. FACS, fluorescence-activated cell sorting.

ARID1A defective cells are exposed to an ATRi, chromosomal rearrangements are observed (Fig. 5d,e). We do note that the potential for ATR inhibition to alter the genomic stability of *ARID1A* mutant tumours in a genotype specific manner (and therefore potentially tumour cell specific manner) might have beneficial therapeutic effects but could also have deleterious effects in driving tumourigenesis or drug resistance by the generation of new oncogenic mutations and chromosomal rearrangements. This is a challenge not just to the use of ATRi but is also an issue of concern when using other agents whose mechanism of action results in changes to the structure of the genome. This issue could be addressed by assessing what additional processes are required to ensure that any increase in genomic instability results in tumour cell death, rather than cell survival with a reordered genome.

Finally, we also note that the precise composition of the BAF complex and the influence of other driver mutations in *ARID1A* mutant tumours might also be important in determining the ARID1A/ATR synthetic lethality and our subsequent work will focus on assessing how the balance between the different members of the SWI/SNF complex might modulate the TOP2A defect and ATRi sensitivity.

## Methods

**Cell lines.** ES2 and TOV21G were obtained from the American Type Tissue Collection. RMG-1, SMOV2, KOC7C, HCH1, OVAS, OVISE, OVMANA, OVTOKO, OVSAYO and KK were courtesy of Dr Hiroaki Itamochi (Tottori University School of Medicine, Yonago, Japan). OCCC lines were grown in McCoys with 10% fetal calf serum (FCS). The identity of cell lines was confirmed by short tandem repeat typing using the StemElite Kit (Promega) in March 2013 and yearly thereafter. ARID1A HCT116 isogenic cell lines were obtained from Horizon Discovery and grown in McCoys with 10% FCS. *Arid1a* null and wild-type mouse ES cells were obtained from Dr Zhong Wang (Harvard Medical School, USA) and grown on gelatin coated plates in DMEM with 10% FCS supplemented with 0.1 mM NEAA, 1 mM sodium pyruvate, 0.1 mM β-mercaptothanol and 2,000 U LIF ml$^{-1}$.

**VE-821 siRNA screen.** A siRNA library (1,280 siRNAs listed in Supplementary Data 1) was purchased from Dharmacon. Genes were selected as described in the main text. Each well contained a SMART pool of four distinct siRNA species targeting different sequences of the target transcript. Each plate was supplemented with negative siCONTROL (12 wells; Dharmacon) and positive control (four wells, siPLK1, Dharmacon). RNAi screening conditions were optimized and raw CellTitre-Glo (Promega) luminescent viability readings were generated as previously described[30]. VE-821 or vehicle (DMSO) was added 24 h after transfection at 1 µM concentration in media and cells were exposed for 5 days. Statistical analysis of the siRNA screen was performed as described elsewhere[30]. In brief, luminescence values from CellTitre-Glo assays in ATRi and DMSO exposed cells were log$_2$ transformed and then normalized to plate median (PM) effects. Drug Effect (DE) scores were calculated from PM normalized data using the equation: DE = (log$_2$ PM normalized signal of siRNA in the presence of ATRi)—(log$_2$ PM normalized signal of siRNA in the absence of ATRi). DE values were then Z-score standardized according to screen median and median absolute deviation values.

**Chemicals.** The ATRi VE-821 and VX-970 were provided by Vertex Pharmaceuticals. Olaparib and BMN673 were purchased from Selleck Chemicals; cisplatin was purchased from Sigma.

**Western blotting and antibodies.** Whole-cell protein extracts were prepared from cells lysed in NET-N buffer (20 mM Tris pH 7.6, 1 mM EDTA, 1% NP40, 150 mM NaCl) supplemented with protease inhibitor cocktail tablets (Roche, West Sussex, UK). Western blots were carried out with precast Bis-Tris gels (Invitrogen, Paisley, UK). The following primary antibodies were used in this study; ARID1A (1:1,000, Cell Signaling, D2A8U), ATR (1:500, Santa Cruz, sc1887), P-T1989 ATR (1:1,000, Gene Tex, GTX128145), Tubulin (1:10,000, Sigma, T6074), γH2AX (1:3,000, Cell Signaling, 2,577), PARP-1 (1:3,000, Santa Cruz, sc-8007), Ezrin (1:3,000, Cell Signaling, 3,145), TOP2A (1:1,000, Cell Signaling, D10G9), Cyclin B1 (1:3,000, Cell Signaling, 4,138), Histone H3 (1:3,000, Cell Signaling, 9,715) and Lamin A/C (1:3,000, Cell Signaling, 4,777). Uncropped scans for all western blots shown in Figs 1–5 are shown in Supplementary Figs 6–10.

**Cellular viability assays.** Short-term survival assays were performed in 384-well plates. Exponentially growing cells were plated at a concentration of 500 cells per well. Drug was added 24 h after seeding and cells were continuously exposed to the drug for 5 days, after which cell viability was estimated using CellTitre-Glo luminescence (Promega). For clonogenic assays, cells were seeded in six-well plates (500 cells per well) and continuously exposed to drug for 14 days. Media containing fresh drug was replaced every 72 h. Cells were fixed with 10% trichloroacetic acid and stained with sulphorhodamine B (Sigma-Aldrich, Gillingham, UK). Colonies were counted manually. All cell-based assays were performed at least in triplicate. Lines of best fit were plotted using a four-parameter nonlinear regression or linear models where appropriate.

**Cell cycle analysis.** Cells were plated at a density of $2 \times 10^5$ cells per well of a six-well plate and incubated for 24 h after which ATRi or 0.1% DMSO was added for the indicated period of time. After incubation, adherent cells were harvested and then fixed with cold 50 (v/v)% ethanol in PBS. Cells were then treated with RNase A for 30 min before nucleic acid staining with propidium iodide (PI, Sigma). Samples were analysed on a BD LSR II flow cytometer using BD FACSDiva software (BD Biosciences). For synchronization experiments cells were incubated with 2 mM thymidine for 12 h, washed and incubated with media for 16 h, then incubated in 2 mM thymidine for a further 12 h. Cells were then released into media containing either VX-970 or DMSO and samples fixed as above for cell cycle analysis.

**p-H3 assay.** Following experimental treatment, cells were fixed using cold 50% ethanol and PBS. Before analysis, cells were centrifuged and resuspended in 1 ml 0.25% (v/v) solution of Triton-X100 in PBS for 15 min. Following centrifugation, cells were resuspended in 100 µl of PBS solution containing 1% (w/v) bovine serum albumin (BSA) and 0.75 µg of P-S10 Histone H3 antibody (Jackson ImmunoResearch). Samples were incubated at room temperature for 3 h. Samples were then centrifuged and washed with PBS solution containing 1% BSA. Cells were subsequently suspended in 100 µl of FITC-labelled goat anti-rabbit secondary antibody (Jackson ImmunoResearch) in a 1:30 dilution in PBS with 1% BSA. Samples were incubated in the dark for 30 min and resuspended in PBS solution containing PI (5 µg ml$^{-1}$) and RNase (1 mg ml$^{-1}$). Samples were analysed on a BD LSR II flow cytometer using BD FACSDiva software (BD Biosciences).

**Apoptosis assay.** HCT116 isogenic and OCCC cells were plated in a 96-well plate at high density (15–20,000 per well). Twenty four hours later serial dilutions of VX-970 were added for the indicated period of time. ApoTox-Glo Triplex Assay (Promega) was performed as per the manufactures protocol.

***In vivo* efficacy studies.** *In vivo* efficacy studies were performed using HCT116 *ARID1A*$^{+/+}$, HCT116 *ARID1A*$^{-/-}$ and TOV21G cells injected subcutaneously in the flank of female CD-1 Nude mice. Animals were treated with either vehicle alone (10% D-α-Tocopherol polyethylene glycol 1000 succinate) or VX-970 (60 mg kg$^{-1}$ in 10% D-α-Tocopherol polyethylene glycol 1000 succinate) by oral gavage. Treatment was administered four times weekly for the indicated length of time. Tumours were measured manually by calliper trice weekly and animals were sacrificed when tumours reached > 15 mm in any direction. All *in vivo* modelling was carried out according to ARRIVE guidelines, regulations set out in the UK Animals (Scientific Procedures) Act 1986, and in line with a UK Home Office approved project licence held by CJL and approved by the ICR ethics board.

**Anaphase bridge analysis.** Cells were grown on Poly-lysine-coated coverslips for 24 h before exposure to the indicated treatment. Samples were fixed in formaldehyde (4%), permeablized in Triton X-100 (0.2%) and DNA stained with DAPI. Slides were then imaged at 60X on a Leica confocal microscope.

**Mitotic spreads.** Following exposure to the indicated treatment, cells were incubated with 0.5% colchicine for 4 h. Cells were harvested, washed in PBS and incubated in 0.56% KCl at 37C for 20 min. Samples were then fixed (3 methanol:1 acetic acid) and DAPI was added. Cell solutions were dropped onto clean coverslips and mitotic spreads imaged at × 60 on a Leica confocal microscope.

**Transcriptomic analaysis.** Whole transcriptome analysis was performed on a Illumina BeadArray HumanHT-12 v4 device at the Wellcome Trust Centre for Human Genetics (University of Oxford). Three biological replicates of HCT116 *ARID1A*$^{+/+}$ and *ARID1A*$^{-/-}$ were analysed. Differential expression and statistical analysis was performed using the LIMMA package from the Bioconductor project[47].

**Immunofluorescence analysis of γH2AX.** Cells were grown on poly-lysine-coated coverslips for 24 h before treatment. Following treatment cells were fixed in 4% paraformaldehyde and permeablized with 0.2% Triton X-100. Cells were stained with antibodies to γH2AX (Millipore, 05–636), Cyclin A (Abcam, ab181591) and DAPI. Fluorescently labelled secondary antibodies were

incubated 30 min before mounting cells on slides. Slides were imaged at ×40 on a Leica confocal microscope.

**Data availability.** All relevant data not presenting in the main figures or Supplementary Data is available from the authors.

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

## Acknowledgements

We thank Dr Zhong Wang (Harvard Medical School, USA) for providing Arid1a null and wild-type mouse ES cells. This work was funded by Breast Cancer Now as part of their funding to the Breast Cancer Now Toby Robins Breast Cancer Research Centre, Cancer Research UK (grant number C347/A8363) and Vertex Pharmaceuticals. We acknowledge NHS funding to the NIHR Royal Marsden Hospital Biomedical Research Centre. CJR is a Sir Henry Wellcome Fellow jointly funded by Science Foundation Ireland, the Health Research Board, and the Wellcome Trust (grant number 103049/Z/13/Z) under the SFI-HRB-Wellcome Trust Biomedical Research Partnership.

## Author contributions

C.T.W., R.M., H.N.P., S.E.J., R.R and R.B. designed, conducted and analysed in vitro cell-based experiments. C.T.W., J.F., A.K., N.B., P.B.V. and A.R.R. conducted and analysed in vivo experiments. C.T.W., R.M., J.C., A.G., C.J.R. analysed high-throughput screen data. P.M.R and J.R.P. developed and provided small molecule inhibitors and advised on their use. A.A. and C.J.L. directed the research and secured funding. All authors contributed to the writing of the manuscript and approved the final version.

## Additional information

**Competing financial interests:** P.M.R. and J.R.P. are paid employees of Vertex Pharmaceuticals. Part of this work has been funded by Vertex Pharmaceuticals as part of a Sponsored Research Agreement between Vertex and The Institute of Cancer Research, London.

