## [Peer Review File · Nature Communications]

Reviewers' comments:

Reviewer #3 (Remarks to the Author):

This is a revised manuscript by Williamson et al which describes a synthetic lethal interaction between ATR and ARID1A, manifest by treatment of ARID1A deficient tumors or cells with ATRi. In this revised manuscript, Williamson et al have sought to address reviewer concerns raised during a prior review, and they have successfully addressed the majority of the issues raised. However, there is still some concern about the author's interpretation that the mechanism underlying the observed synthetic lethality is based primarily or solely on the decatenation defect resulting from ARID1A loss. Indeed, their new data strongly suggests that there is a defect in S phase in these cells and that this may contribute to the synthetic lethality with ATRi. I think this work would be very appropriate for Nature Communications, provided this alternative explanation is addressed as outlined below.

Fig 5A: These data, specifically at the 3, 4 and 5 hour time points, suggest that the ARID1A $-/-$ cells are delayed in S phase compared to ARID1A $+/+$ cells (irrespective of ATR inh). This finding argues against the interpretation that this is a result of ARID1A effects on Topo loading (at least as the sole result). In fact, it suggests that ARID1a mutations lead to S phase defects (that is in DNA synthesis and not decatenation) which could lead to a dependency on ATR for survival. This question was specifically addressed in the revised manuscript in Figure 4B, however the time course was not detailed enough and this effect was lost (only 2, 6 h were provided in Figure 4B and 3-5h timepoints were excluded). This point needs to be made and discussed. I suggest they use the data in Figure 5A to address this, and remove the Figure 4B data as it is misleading without the 3-5h timepoints.

Following on this point, the authors should also look at H2AX phosphorylation in S phase cells as initially suggested, to further test this idea. This could be done by western in synchronized cells at the 3, 4, 5 h time points, or preferably by microscopy on single cells, where H2AX phosphorylation could be assessed specifically in the S phase cells (judged by DNA content).

Minor Point

In Fig 5D and E - the chart suggest ~3 breaks in the $-/-$ cells with inhibitor, whereas the image shows massive fragmentation. A more representative image should be utilized.

Fig 5A is very difficult to read. I suggest they add some space between time points to facilitate analysis.

The words in the model are difficult to read (in the top part, showing the BAF complex).

Reviewer #4 (Remarks to the Author):

ATR inhibitors as a Synthetic Lethal Therapy for Tumors Deficient in ARID1A
Chris T. Williamson et al

The authors have effectively responded to my own as well as the other reviewers suggestions with

new studies and rewriting. The studies described in the manuscript will lay the groundwork for the first potentially therapeutic approach to the large group of tumors that have mutations in BAF complex subunits. The studies are quite extensive and include animal models, xenographs and a depth of evidence that paves the way for therapeutic studies.

A minor point:

In Figure 4 the authors show that Arid1a^{-/-} cells have reduced H3P staining but a G2/M arrest, which they take to be indicative of a decatenation defect. The authors discuss this on line 390. Could it be that top2 inhibition or Arid1a ko causes a reduction in the transcriptional functions of Top2 as well and hence leads to an earlier block than the Brg deletion phenotype?

Reviewers' comments:**Reviewer #3 (Remarks to the Author):**

This is a revised manuscript by Williamson et al which describes a synthetic lethal interaction between ATR and ARID1A, manifest by treatment of ARID1A deficient tumors or cells with ATRi. In this revised manuscript, Williamson et al have sought to address reviewer concerns raised during a prior review, and they have successfully addressed the majority of the issues raised. However, there is still some concern about the author's interpretation that the mechanism underlying the observed synthetic lethality is based primarily or solely on the decatenation defect resulting from ARID1A loss. Indeed, their new data strongly suggests that there is a defect in S phase in these cells and that this may contribute to the synthetic lethality with ATRi. I think this work would be very appropriate for Nature Communications, provided this alternative explanation is addressed as outlined below.

Our response:

We thank the reviewer for taking the time to look at our revised manuscript.

In a revised version of the manuscript, we have now highlighted the possibility that a decatenation defect might not be the only cause of the ARID1A/ATR synthetic lethality and that other mechanisms could also contribute to this phenotype. In our subsequent comments (below), we detail how this has been addressed.

Reviewer 3 continued:

Fig 5A: These data, specifically at the 3, 4 and 5 hour time points, suggest that the ARID1A ^{-/-} cells are delayed in S phase compared to ARID1A ^{+/+} cells (irrespective of ATR inh). This finding argues against the interpretation that this is a result of ARID1A effects on Topo loading (at least as the sole result). In fact, it suggests that ARID1a mutations lead to S phase defects (that is in DNA synthesis and not decatenation) which could lead to a dependency on ATR for survival. This question was specifically addressed in the revised manuscript in Figure 4B, however the time course was not detailed enough and this effect was lost (only 2, 6 h were provided in Figure 4B and 3-5h timepoints were excluded). This point needs to be made and discussed. I suggest they use the data in Figure 5A to address this, and remove the Figure 4B data as it is misleading without the 3-5h timepoints.

Our response:

We agree with the reviewer's interpretation of the data that ARID1A^{-/-} cells demonstrate a slower transition through S phase in addition to a slower transition through G2 into mitosis.

We have revised the cell cycle analysis of ARID1A wild type and ARID1A^{-/-} cells and now show in a new Figure 4B an hourly analysis of cell cycle progression over 10 hours after release from double thymidine block:

Figure 4B- Histogram of the cellular DAPI-stained DNA content, determined by FACS, in HCT116 ARID1A^{+/+} and ARID1A^{-/-} cells at the indicated time points following release from double thymidine synchronisation in G₁/early S phase.

This analysis indicates that, as the referee suggested, that there is a slower progression of ARID1A^{-/-} cells through S phase as well as a reduced progression from G₂ into mitosis. We have now modified the main text of the manuscript as follows:

“To investigate this further, we synchronised cells at the G₁/early S-phase boundary using a double thymidine block and then followed cell cycle progression once the thymidine block was removed (Figure 4B). We found that, compared to ARID1A^{+/+} cells, ARID1A^{-/-} cells had a delayed progression through S phase and a slower progression from G₂ into mitosis (8-10 hours after block - Figure 4B).”

Reviewer 3 continued:

Following on this point, the authors should also look at H2AX phosphorylation in S phase cells as initially suggested, to further test this idea. This could be done by western in synchronized cells at the 3, 4, 5 h time points, or preferably by microscopy on single cells, where H2AX phosphorylation could be assessed specifically in the S phase cells (judged by DNA content).

Our response:

Following the reviewer's suggestion, we have also performed two additional experiments. Firstly, we synchronized ARID1A^{+/+} and ARID1A^{-/-} cells in G₁/early S phase using double thymidine block and released these into the cell cycle in the presence or absence of the ATR inhibitor VX-970 for five hours, whilst the cells were in S phase (new Supplementary Figure 5E). Western blot analysis suggested that there was an increase in γ H2AX levels in ARID1A^{-/-} cells by five hours after removal of the double thymidine block:

Supplementary Figure 5E- Western blot of cell lysates taken from the indicated time points from HCT116 ARID1A^{+/+} and ARID1A^{-/-} cells following release from synchronization in G₁/early S phase by double thymidine block. Cells were released from thymidine block into media containing DMSO or VX-970 (0.5 μ M).

This suggested that some of the DNA damage associated with ATR inhibitor exposure in ARID1A^{-/-} cells could be occurring in S phase.

To investigate this further, we used confocal microscopy to examine γ H2AX levels in cells exposed to ATR inhibitor for a short, two hour, period. In this experiment we also examined Cyclin A expression so that we could distinguish cells in G₁ (Cyclin A negative) from cells in S/G₂ (Cyclin A positive). We found that both Cyclin A positive and negative ARID1A^{-/-} cells exhibited an increase in γ H2AX immunofluorescent staining when exposed to ATR inhibitor (new Supplementary Figure 5A-D):

(legend overleaf)

Supplementary Figure 5A-D

A. Representative immunofluorescence images of HCT116 ARID1A^{+/+} and ARID1A^{-/-} cells stained with DAPI, Cyclin A, γ H2AX and a merged image, following treatment with DMSO, IR (10 Gy, 2 hours) and VX-970 (0.5 μ M, 2 hours). Scale bar equals 75 μ M. **B.** Fraction of Cyclin A negative staining positive for γ H2AX (>10 foci and/or pan nuclear staining) following DMSO or VX-970 (0.5 μ M, 2 hours) treatment in HCT116 ARID1A^{+/+} and ARID1A^{-/-}. Error bars represent SD for 3 biological replicates. No less than 200 cells were counted on four large field images for each replicate. P-values calculated using Student's t-test and * indicates statistical significance ($p < 0.05$). **C.** Fraction of Cyclin A positive staining positive for γ H2AX (>10 foci and/or pan nuclear staining) following DMSO or VX-970 (0.5 μ M, 2 hours) treatment in HCT116 ARID1A^{+/+} and ARID1A^{-/-}. Error bars represent SD for 3 biological replicates. No less than 200 cells were counted on four large field images for each replicate. P-values calculated using Student's t-test and * indicates statistical significance ($p < 0.05$). NS indicates the comparison is not statistically significant ($p > 0.05$ by Student's t-test). **D.** Representative images of the various classifications used to score cells as H2AX positive in panels B and C. Scale bar equals 6 μ M.

Taken together, this suggested that the γ H2AX response to ATR inhibitor exposed ARID1A^{-/-} cells was not restricted to S phase but also apparent in cells in G₁. Given the short exposure to ATR inhibitor (2 hours) it is possible that the γ H2AX response in G₁ could be initiated in G₁ itself or could be the result of DNA damage occurring during mitosis and apparent as cells transit into G₁ during the two hour period.

On the basis of these results, we have now modified the main text as follows:

"In ARID1A^{-/-} cells exposed to ATRi for a short period (two hours), we found an increase in H2AX phosphorylation in Cyclin A positive as well as negative cells (Supplementary Figure 5A-D). The presence of VX-970-induced H2AX phosphorylation in S phase cells was confirmed by western blotting in synchronised cells as they transitioned through S phase (Supplementary Figure 5E). Taken together, this suggested that the γ H2AX response to ATRi in ARID1A^{-/-} cells occurred in S/G₂ phases (Cyclin A positive) as well as in G₁ phase (Cyclin A negative) of the cell cycle, implying that processes occurring in multiple phases of the cell cycle could contribute to the synthetic lethal phenotype observed. It is possible that the γ H2AX response in Cyclin A negative cells could be initiated in G₁ itself or could be the result of DNA damage occurring during mitosis and apparent as cells transit into G₁ during the two hour ATR inhibitor exposure."

We have also modified our model (Figure 5I) to include the possibility that ATR inhibitors are able to selectively target ARID1A-deficient cells as a result of both a cell cycle progression defect as well as the TOP2A decatenation defect. The proposed model is presented in the results section

as follows:

“From this data we conclude that loss of ARID1A function results in: (i) a defect in the ability of cells to recruit TOP2A to chromatin and (ii) cell cycle progression defects in both S and G₂/M phases of the cell cycle. It seems possible that these factors combined or in isolation might render tumour cells sensitive to small molecule ATR inhibitors as these agents impair the ability of cells to mount adequate DNA damage responses, whilst at the same time accelerating mitotic entry (Figure 5).”

In addition we have modified our model Figure 5I:

Figure 5I:

A model for the proposed mechanism driving the sensitivity of ARID1A-deficient cells to ATRi. Loss of ARID1A function results in: (i) a defect in the ability of cells to recruit TOP2A to chromatin and (ii) cell cycle progression defects in both S and G₂/M phases of the cell cycle. These factors combined or in isolation might render tumour cells sensitive to small molecule ATR inhibitors as these agents impair the ability of cells to mount adequate DNA damage responses, whilst at the same time accelerating mitotic entry.

Reviewer 3 continued:

Minor Point

In Fig 5D and E - the chart suggest ~3 breaks in the $-/-$ cells with inhibitor, whereas the image shows massive fragmentation. A more representative image should be utilized.

Our response:

We have now changed Figure 5D so that the image of VX-970 treated HCT116 ARID1A cell is more representative of the quantification in Figure 5E.

Figure 5D- Images of mitotic spreads from HCT116 ARID1A^{+/+} and ARID1A^{-/-} cells following exposure to either DMSO or VX-970 (1 μ M). Scale bar represents 20 μ m.

Reviewer 3 continued:

Fig 5A is very difficult to read. I suggest they add some space between time points to facilitate analysis.

Our response:

We have now modified Figure 5A so that there is more space between the time points to make the data easier to analyze.

Figure 5A- Histogram of the cellular DAPI-stained DNA content, determined by FACS, in HCT116 ARID1A^{+/+} and ARID1A^{-/-} cells at the indicated time points following release from synchronization in G₁/early S phase by double thymidine block. Cells were released from thymidine block into media containing DMSO (blue) or VX-970 (0.5 μM, red).

Reviewer 3 continued:

The words in the model are difficult to read (in the top part, showing the BAF complex).

Our response:

We have now modified our model Figure 5I (see response above) including changing the locations of the text labels to make it easier to read.

Reviewer #4 (Remarks to the Author):

ATR inhibitors as a Synthetic Lethal Therapy for Tumors Deficient in ARID1A
Chris T. Williamson et al

The authors have effectively responded to my own as well as the other reviewers suggestions with new studies and rewriting. The studies described in the manuscript will lay the groundwork for the first potentially therapeutic approach to the large group of tumors that have mutations in BAF complex subunits. The studies are quite extensive and include animal models, xenographs and a depth of evidence that paves the way for therapeutic studies.

A minor point:

In Figure 4 the authors show that Arid1a^{-/-} cells have reduced H3P staining but a G2/M arrest, which they take to be indicative of a decatenation defect. The authors discuss this on line 390. Could it be that top2 inhibition or Arid1a ko causes a reduction in the transcriptional functions of Top2 as well and hence leads to an earlier block than the Brg deletion phenotype?

Our response:

As the reviewer correctly points out TOP2A has been shown to play a key role in transcription, in addition to its function in bulk DNA decatenation after replication and this transcriptional effect could indeed contribute to the synthetic lethal interaction between ARID1A and ATR. To reflect this possibility, we have now added the section below to the discussion section of our revised manuscript:

“Topoisomerase function has also been implicated in the prevention of deleterious collisions between the transcriptional and replication machineries (44), potentially providing another mechanism that if defective, could lead to a dependency upon ATR, and could explain the S phase delay in ARID1A^{-/-} cells. Finally it is possible that loss of ARID1A, by causing alterations in chromatin structure that lead to an altered transcriptional programme, could impair the expression of other genes critical for cells to survive in the face of ATR inhibition.”

REVIEWERS' COMMENTS:

Reviewer #4 (Remarks to the Author):

The reviewers have addressed all my concerns and I think this is an exciting manuscript ready for publication.

REVIEWERS' COMMENTS:

Reviewer #4 (Remarks to the Author):

The reviewers have addressed all my concerns and I think this is an exciting manuscript ready for publication.

Our response: *We thank the reviewer for taking the time to read our revised manuscript. We are glad that we have successfully responded to all of the reviewers concerns.*